# Interventions and strategies involving primary healthcare professionals to manage emergency department overcrowding: a scoping review

Maya M Jeyaraman [1,2] Leslie Copstein,[1] Nameer Al-Yousif,[1] Rachel N Alder,[3] Scott W Kirkland,[4] Yahya Al-Yousif,[1] Roger Suss,[5] Ryan Zarychanski,[2,6] Malcolm B Doupe [2] Simon Berthelot,[7] Jean Mireault,[8] Patrick Tardif,[9] Nicole Askin,[10] Tamara Buchel,[11] Rasheda Rabbani,[1,2] Thomas Beaudry,[12] Melissa Hartwell,[13] Carolyn Shimmin,[1] Jeanette Edwards,[14] Gayle Halas [15] William Sevcik,[4] Andrea C Tricco [16] Alecs Chochinov,[17] Brian H Rowe,[4,18] Ahmed M Abou-Setta[1,2]

► Prepublication history and supplemental material for this paper is available online. To view these files, please visit the journal online (http://dx.doi.org/10.1136/bmjopen-2021-048613).

For numbered affiliations see end of article.

**Correspondence to**
Dr Maya M Jeyaraman;
maya.jeyaraman@umanitoba.ca

## ABSTRACT

**Objectives** To conduct a scoping review to identify and summarise the existing literature on interventions involving primary healthcare professionals to manage emergency department (ED) overcrowding.

**Design** A scoping review.

**Data sources** A comprehensive database search of Medline (Ovid), EMBASE (Ovid), Cochrane Library (Wiley) and CINAHL (EBSCO) databases was conducted (inception until January 2020) using peer-reviewed search strategies, complemented by a search of grey literature sources.

**Eligibility criteria** Interventions and strategies involving primary healthcare professionals (PHCPs: general practitioners (GPs), nurse practitioners (NPs) or nurses with expanded role) to manage ED overcrowding.

**Methods** We engaged and collaborated, with 13 patient partners during the design and conduct stages of this review. We conducted this review using the JBI guidelines. Two reviewers independently selected studies and extracted data. We conducted descriptive analysis of the included studies (frequencies and percentages).

**Results** From 23 947 records identified, we included 268 studies published between 1981 and 2020. The majority (58%) of studies were conducted in North America and were predominantly cohort studies (42%). The reported interventions were either 'within ED' (48%) interventions (eg, PHCP-led ED triage or fast track) or 'outside ED' interventions (52%) (eg, after-hours GP clinic and GP cooperatives). PHCPs involved in the interventions were: GP (32%), NP (26%), nurses with expanded role (16%) and combinations of the PHCPs (42%). The 'within ED' and 'outside ED' interventions reported outcomes on patient flow and ED utilisation, respectively.

**Conclusions** We identified many interventions involving PHCPs that predominantly reported a positive impact on ED utilisation/patient flow metrics. Future research needs to focus on conducting well-designed randomized controlled trials (RCTs) and systematic reviews to evaluate the effectiveness of specific interventions involving PHCPs

## Strengths and limitations of this study

► A major strength of our study is that we collaborated with 12 patient partners during the design, conduct and dissemination stages of this scoping review to refine the review question, identify patient-important review outcomes, develop search terms, grey literature search and knowledge dissemination.

► This comprehensive scoping review was conducted using JBI guidelines and an a priori registered protocol without any restrictions on publication dates, population or study design.

► A limitation of this scoping review is that we only included English-language publications.

to critically appraise and summarise evidence on this topic.

## INTRODUCTION

Emergency department (ED) overcrowding is an increasing global crisis,[1] leading to challenges in the delivery of timely healthcare to patients, which can result in adverse patient outcomes.[2 3] Overcrowding occurs when the demand for services exceeds the ability of the ED to provide quality care within acceptable time frames.[4] Studies have shown how ED crowding compromises patient safety and contributes to poor quality of care, such as delays in antibiotic treatment,[5] poor pain management for patients with severe pain,[6] increased patient mortality[7] and staff frustration.[8 9]

The problem of ED overcrowding is complex and challenging, and models have been proposed to characterise it using input

(eg, health lines, patient volume and diversion strategies), throughput (eg, provider availability, diagnostic testing and response to therapy), output (eg, consultation times, ED boarding and lack of hospital beds) and system-wide (eg, remuneration, pay-for performance, public reporting and accountability frameworks) factors.[10] One such input factor impacting ED overcrowding is the large volume of low-acuity patient visits to the ED, which can stress the available resources in the ED, leading to decreased patient and staff satisfaction.[11 12]

A lack of access to a primary care provider (PCP),[12–14] as well as an inability to see their PCP after-hours,[15] or to get an appointment within an appropriate time frame[3 15 16] can often force patients with low-acuity conditions to turn to ED as their last resort.[17] Some have suggested that many of these presentations represent primary care sentinel conditions (eg, asthma, hypertension and low back pain) which could be better managed through greater continuity and ongoing follow-up from PCPs.[18] In many rural communities, EDs are often the only source of primary care.[4]

As such, it is crucial to identify interventions and strategies involving primary healthcare professionals (PHCP) that have an impact on the ED metrics. Using a scoping review methodology, in order to map the existing literature on this topic, the main objective was to identify and summarise existing literature on the interventions and strategies involving PHCPs (family physicians/general practitioners (GPs), nurse practitioners (NPs) or nurses with expanded role) to manage ED overcrowding.

## METHODS
An a priori protocol was developed and posted on the Open Science Framework platform.[19] This scoping review was conducted using the methodologically rigorous JBI guideline for scoping reviews,[20] and the six-stage methodological framework outlined by Levac *et al.*[21] This scoping review adheres to the reporting guidelines of the Preferred Reporting Items for Systematic reviews and Meta-Analyses extension for Scoping Reviews (PRISMA-ScR).[22 23]

### Population, concept and context
Studies were included in the review if they investigated an intervention or strategy to manage ED overcrowding and ED patient flow and involved a PHCP, including family physician/GP, NP or nurse with increased authority. Editorials, commentaries, reviews and historical articles were excluded from the review. For feasibility, only English-language publications were included. Additional details regarding study eligibility criteria (population, concept and context) are available in online supplemental appendix table 1.

### Search methods for identifying relevant citations
A health librarian (TR) with experience conducting systematic literature searches designed a comprehensive search strategy for Medline (Ovid) to identify literature relevant to the objectives (online supplemental appendix table 2). After feedback from the principal investigators and the patient partners, the Medline search was then peer reviewed by an independent librarian (JJ) using the Peer Review of Electronic Search Strategies (PRESS) checklist.[24] Once finalised, it was adapted for use in the following three additional electronic databases: EMBASE (Ovid), Cochrane Library (Wiley) and CINAHL (EBSCO). The databases were searched from their inception to January 2020 by an experienced librarian (NA). A search of several grey literature sources (online supplemental appendix table 3) was conducted to identify additional publications. Reference lists of all the included publications were searched for additional relevant studies. References were imported and managed in EndNote (V.X8, Thomson Reuters, New York, New York, USA).

### Selection of sources of evidence
Two reviewers (NA-Y and (LC, YA-Y or RA)) independently reviewed and screened the titles and abstracts of all citations identified by our search strategy using standardised pilot-tested (n=10) screening forms. Citations identified as potentially relevant, or which the reviewers were unclear of its inclusion were eligible for full-text screening. Two reviewers (NA-Y and (LC or YA-Y)) independently reviewed full texts of all included studies and discrepancies were resolved via consensus or a third-party adjudicator (MJ).

### Charting information, risk of bias assessment and synthesis of results
A standardised data extraction form was developed, and pilot tested (n=10) independently by reviewers in the team. As a part of data extraction, two reviewers (LC and (NA-Y, YA-Y or RA)) independently extracted data from the included studies. When required, the form was modified, using an iterative consensus-based process.[25] All disagreements among reviewers were resolved by consensus and checked for accuracy by a third reviewer (SK and/or MJ). Extracted data included: study identification, author(s), publication year, publication type, study design, country of origin, type of ED (urban vs rural; adult, paediatric or mixed), type of intervention or strategy (within ED or outside ED), type of PHCP (GP, NP or nurse with increased authority), nature of intervention or strategy (added a PHCP, or increased responsibility to a PHCP) and the nature of impact of interventions on ED-related outcomes as reported by the study authors in the included studies (positive impact, negative impact or no impact).

Consistent with established scoping review methodology,[25 26] we did not appraise the risk of bias of the included studies, nor did we summarise the data quantitatively (meta-analyses). Instead, we synthesised the findings using descriptive statistical analyses (eg, frequencies and percentages) of the extracted variables, and reported using graphs and tables.

## Patient and public involvement (collaboration with stakeholders)

We collaborated with a diverse group of 13 patient partners (self-identified as indigenous, immigrant, white and/or living with disability) identified with the help of Strategy for Patient-Oriented Research (SPOR) Support for People and Patient-Oriented Research and Trials units (Manitoba, Alberta and Quebec), during the design phase, grant application phase and in the conduct of this review. The patient engagement process has been reported according to the Guidance for Reporting Involvement of Patients and the Public (GRIPP2) checklist (short form).[27] During the design phase, 12 patient partners collaborated with the principle investigators (12 one-on-one interviews followed by discussion group to arrive at a consensus) to refine the review question to focus on interventions involving PHCPs, to finalise review inclusion criteria, and to select patient-important outcomes. During the grant application phase, they supported our grant application as knowledge users and provided letters of support. While conducting this scoping review, three patient partners collaborated with researchers in refining the search strategy, reviewing the included studies, identifying grey literature, interpreting study results (we shared knowledge of our study findings and integrated information obtained from them in the reporting of the study findings and future directions) and in the knowledge dissemination process (conference presentation). Patient partners were compensated for their time.

Our team members (researchers and patient partners) from multiple provinces across Canada, and the partner organisations provided ongoing feedback. Three presentations were made to various members of the team during the conduct of this review by the principal investigator to discuss the extracted data, obtain feedback regarding the inclusion and classification of various identified interventions and strategies, as well as review outcomes of importance, and to discuss the future directions of the project (discussions regarding patient-important research priorities and review questions for future systematic reviews). Partnering organisations included Manitoba Medical Service Foundation, SPOR Support units (Manitoba, Alberta and Quebec), Emergency Medicine Research Group, and Primary and Integrated Healthcare Innovation network, and the knowledge users (Winnipeg Regional Health Authority, Emergency Strategic Clinical Network, Shared Health Manitoba and Manitoba College of Family Physicians).

## RESULTS

Our literature search identified 23 947 records from the initial database search, of which 268 studies (274 reports) met our inclusion criteria. A detailed description of the study selection process for this scoping review is depicted using the PRISMA study flow diagram for scoping reviews[22 28] (figure 1).

## Main results

### Study characteristics

The included studies (online supplemental appendix table 4) were published between 1981 and 2020 with the majority of studies published in the last 10 years (2010–2020), showing an upward trend in the number of publications per year over time (1981–2020; figure 2). For details regarding the key characteristics of the included studies, see table 1 and online supplemental appendix table 5. The included studies were mostly cohort (prospective and retrospective) studies (41.8%) from North America (58%) and Europe (28.4%) and published in peer-reviewed journals (76.9%). Nearly half of the included studies were conducted in USA (47%) and only 8.2% were randomised controlled trials. The ED location was most often urban (n=137, 51%) and 16 studies (6%) reported an ED location that provided care to patients from rural and urban areas. Most of the studies did not specify the type of ED (n=146, 54.5%); however, mixed (n=79, 29.5%), adult (n=19, 7%) or paediatric (n=24, 9%) ED types were documented.

### Interventions and strategies involving PHCPs

For this review, interventions or strategies involving PHCPs were categorised as either 'within ED' or 'outside ED' interventions. A total of 139 studies (52%) reported 'outside ED' interventions or strategies, and 129 studies (48%) reported 'within ED' interventions or strategies (online supplemental appendix table 6). The study intervention for each of the 268 included studies has been reported in online supplemental appendix table 7.

Several 'within ED' interventions and strategies identified in this review. These interventions were grouped under four common themes as follows: (1) area within ED staffed by PHCPs to manage lower acuity ED patients streamlined at triage,[29] (2) PHCPs located next to ED (sharing common triage with ED) to manage lower acuity ED patients streamed at triage as well as self-directed patients,[30] (3) PHCPs located at ED triage[31] to manage lower acuity ED patients and (4) PHCPs fully integrated within the ED to manage ED patients along with the ED team[32] (online supplemental appendix table 6).

The PHCP triage intervention involved patient management strategies such as, 'see and treat' lower acuity patients,[31] diverting low-acuity patients to adjacent/co-located primary care centre or after-hours primary care centre,[33] or the triage nurse was given increased authority to order diagnostic investigations or initiate a specified protocol.[34] A few 'within ED' interventions involved low-acuity patients streamlined at triage to a PHCP working alone within ED (eg, rapid medical assessment units,[35] fast-track units[36] or emergency care access points[35] for management). Other 'within ED' interventions (non-triage) involved PHCPs working in conjunction with the emergency physician and the rest of the ED team in the interpretation of diagnostic imaging, management of lower acuity or in some cases higher acuity patients, and in discharge process.[32]

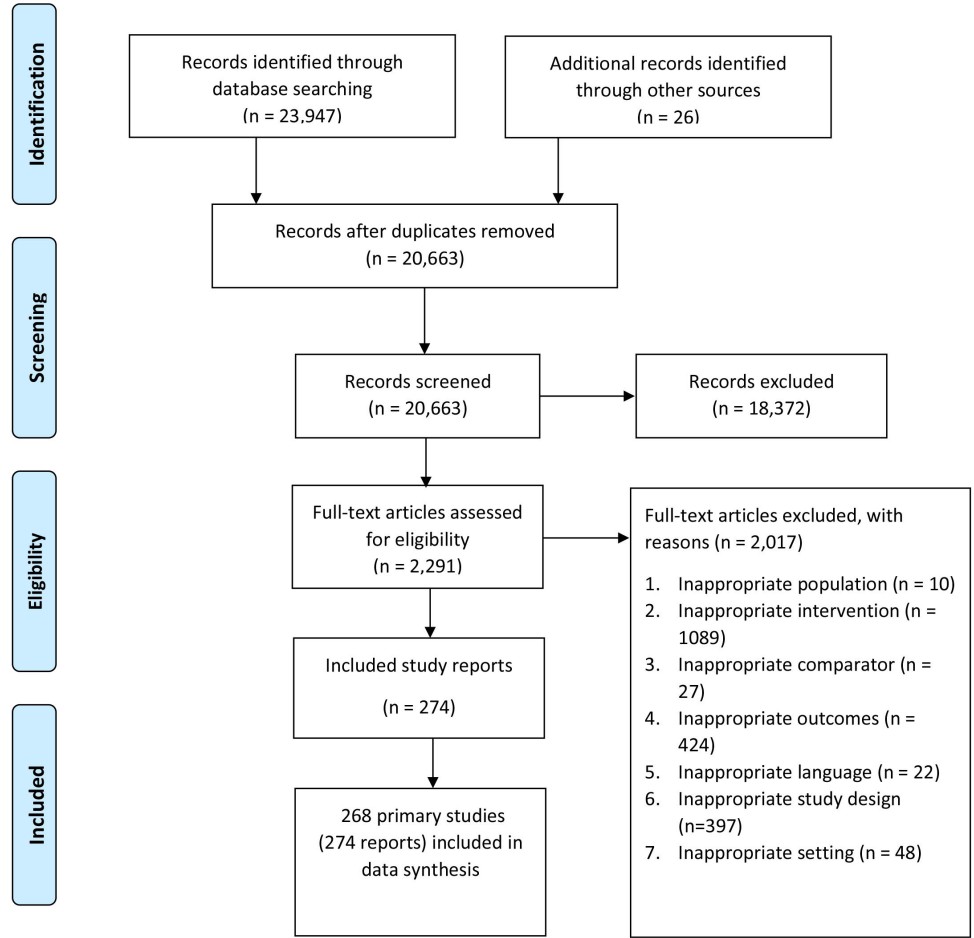

**Figure 1** Preferred Reporting Items for Systematic reviews and Meta-Analyses study flow chart for scoping review of primary healthcare professional interventions to address emergency department overcrowding.

Studies employed several 'outside ED' interventions or strategies. We identified four common themes across these interventions and grouped them under these following themes: (1) improving timely access to primary care, (2) integrating hospital and GP care, (3) providing financial support and (4) implementing new clinics/

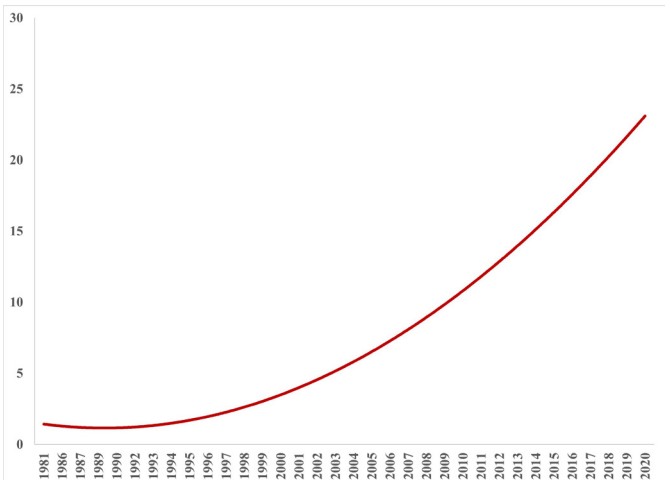

**Figure 2** Number of publications per year involving primary healthcare professional interventions to address emergency department overcrowding.

services (online supplemental appendix table 6). Some examples of the 'outside ED' interventions or strategies were: increasing after-hours primary care,[37] free access to primary care for the uninsured,[38] adjacent or co-located primary care clinic for lower acuity patients,[39] introduction of a patient-centred medical home that addresses primary care needs of patients,[40] implementation of GP cooperatives (out-of-hours primary healthcare in one centrally located practice),[41] urgent care collaborations between the GP and ED,[42] GP-led walk-in centres,[43] PCP blended fee for service,[44] hospital-integrated general practice for emergency care services,[45] integrated emergency posts where care is provided by both ED and GPs,[46] rural health clinics[47] and advanced access primary care with timely access.[48]

### Involvement of primary healthcare providers

The type of PHCP involved in the interventions or strategies involving PHCPs were as follows: family physician (n=85, 31.7%), NP (n=69, 25.7%) or nurse given increased authority (n=43, 16%). There were also a number of dyads reported, including FP and NP (n=15, 5.6%), FP and nurse (n=2, 0.8%), or NP and nurse (n=8, 3%). In some studies, the intervention or strategy was described as primary care with no mention of the specific

**Table 1** Characteristics of included studies in a scoping review of PHCP interventions to address ED overcrowding

| CharacteristicNs | | n (%) (N=268) |
|---|---|---|
| Publication type | Journal article | 206 (76.9) |
| | Abstract | 62 (23.1) |
| Continents | North America | 156 (58.2) |
| | Europe | 76 (28.4) |
| | Oceania | 22 (8.2) |
| | Asia | 9 (3.4) |
| | Middle east | 3 (1.0) |
| | South America | 1 (0.4) |
| | Not reported | 1 (0.4) |
| Type of ED (rural vs urban) | Rural ED | 5 (2.0) |
| | Urban ED | 137 (51.0) |
| | Both | 16 (6.0) |
| | Not reported | 110 (41.0) |
| Type of ED (adult vs other) | Adult ED | 19 (7.0) |
| | Paediatric ED | 24 (9.0) |
| | Mixed ED | 79 (29.5) |
| | Not reported | 146 (54.5) |
| Type of intervention or strategy | Within ED | 129 (48.0) |
| | Outside ED | 139 (52.0) |
| Type of PHCP | FP | 85 (31.7) |
| | NP | 69 (25.7) |
| | Nurse with expanded role | 43 (16.0) |
| | FP and NP | 15 (5.6) |
| | FP and nurse | 2 (0.8) |
| | NP and nurse | 8 (3.0) |
| | Not reported | 46 (17.2) |
| Nature of intervention or strategy | Added a PHCP | 190 (71.0) |
| | Increased responsibility | 59 (22.0) |
| | Not reported | 19 (7.0) |
| Study designs | Randomised clinical trial | 22 (8.2) |
| | Non-randomised trial | 9 (3.4) |
| | Cohort study (prospective or retrospective) | 112 (41.8) |
| | Case–control study | 4 (1.5) |
| | Cross-sectional study | 27 (10.0) |
| | Pre–post study | 76 (28.4) |
| | Interrupted time series | 14 (5.2) |
| | Mixed-methods study | 4 (1.5) |

ED, emergency department; FP, family physician; NP, nurse practitioner; PHCP, primary healthcare professional.

type of PHCP involved (n=46, 17.2%). The majority of studies introduced or added a PHCP (n=190, 70.1%) as a part of the intervention or strategy, and the rest of the

studies increased the responsibility of the PHCPs as a part of the intervention or strategy (n=59, 22%). In a few studies (n=19, 7%), it was not clear if there was an addition or increased responsibility of a PHCP.

### Reported impact of PHCP interventions and strategies on ED outcomes

Several ED-related outcomes, and the impact (positive, negative or no impact) of the PHCP interventions and strategies on these ED outcomes were reported by the included studies. The majority of the included studies that reported 'outside ED' interventions or strategies investigated the impact of these interventions on 'input'-related outcomes (ie, minimising the ED utilisation), whereas studies reporting 'within ED' interventions or strategies investigated the impact of the interventions on 'through-put'-related outcomes (ie, ED patient flow metrics). The majority of the included studies reported ED outcomes such as ED visits, ED length of stay (LOS), leave without being seen (LWBS), patient satisfaction, patient safety, ED workup time, time to provider initial assessment, number of patients diverted to primary care and ED cost savings (online supplemental appendix table 8).

The reported impact (positive impact, negative impact or no impact) of various PHCP-led interventions and strategies on various ED outcomes is reported in online supplemental appendix table 8. About 62% of the included studies reported a positive impact on ED utilisation (ie, decrease in lower acuity ED visits), whereas 28% reported no impact and 9.7% reported negative impact (increased ED utilisation). Similarly, the majority of the included studies reported a positive impact on many important ED-related outcomes such as ED LOS (73.6%), LWBS (77.1%), patient satisfaction (62.1%), time to provider initial assessment (84.6%), ED workup time (62.5%), leave against medical advice (50%), patient safety (100%) and leave before completion of service (50%). The rest of the included studies reported either a negative impact or no impact for the above outcomes.

### DISCUSSION

This scoping review identified 268 unique comparative studies that reported on various interventions or strategies involving PHCPs to manage ED overcrowding and ED patient flow (regardless of the geographic area or the type of healthcare system). Most of the included studies were journal articles of cohort design, published in North America, and were conducted in urban EDs. The interventions or strategies involving PHCPs were implemented either 'within ED' or 'outside ED'.

To the best of our knowledge, this is the first scoping review to describe various interventions or strategies involving PHCPs and their reported impact on ED outcomes. In this scoping review, we identified and described various types of 'within ED' (triage or non-triage) PHCP-led interventions from existing literature. In 2010, Carson et al[19] had described four models in

which lower acuity patients could be managed within ED by PHCPs, as follows: (1) streaming ED patients at triage to an area within ED staffed by PHCPs, for management;[49] (2) PHCPs located next to ED (sharing triage with ED) for management of redirected low-acuity ED patients or self-directed patients;[49] (3) PHCPs located at triage who either 'see and treat' or redirect low-acuity patients to primary care;[49] (4) PHCPs fully integrated with the ED team supporting management of low-acuity or higher acuity patients.[49] In this review, each of the 'within ED' PHCP-led interventions reported by the included studies fit at least one of the four of the Carson's models. It is important to note that the PHCPs in these interventions were providing episodic care (episode-related primary care) and not care continuity.

Although we are unaware of any previous scoping reviews on this topic, there are a few systematic reviews that have been previously published, which found a positive impact of specific PHCP-led 'within ED' interventions on ED patient flow metrics. In 2011, Rowe *et al*[50] reported that triage nurse ordering (ie, triage nurse given increased authority) had a positive impact on ED patient flow metrics. Similarly, in another systematic review, Rowe *et al*[51] had reported a positive impact of triage liaison physicians on ED patient flow metrics. A more recent review reported that NPs practicing in the ED improved ED quality metrics and patient satisfaction,[52] while another review investigating strategies to alleviate access block and overcrowding, suggested that co-locating primary care, fast-track and NPs within ED may have a positive impact.[53] Other reviews, however, have reported a limited impact of 'within-ED' interventions on ED patient flow metrics. A previous Cochrane review,[54] which included only four studies, reported a very low certainty of evidence on the role of GPs and NPs within the ED (but not at triage) on ED flow metrics. In 2015, a narrative review[55] was published on the NP role in the ED and reported that high-quality research is required to demonstrate clinical and service effectiveness.

This review identified numerous types of 'outside ED' interventions that have been employed across the literature. It is important to note that studies reporting 'outside ED' interventions and strategies focused predominantly on reduction of ED utilisation by patients and reported 'ED visits' as an outcome (input-related outcomes) in their study report. Although we are unaware of any scoping reviews on 'outside ED' PHCP-led interventions, there have been a few focused systematic reviews published to date on specific areas under this broad topic. Ansell *et al*[56] summarised interventions to reduce wait times for primary care appointments to mitigate ED utilisation, and concluded that open access scheduling and patient-centred interventions may reduce wait times. Patients seeking ED care due to lack of timely primary care access (same-day or after-hours appointments) may contribute to ED overcrowding and in this context some of the 'outside ED' interventions summarised in this scoping review may be beneficial in mitigating ED overcrowding.

The results of this scoping review found that the majority of studies reported a positive impact of PHCP-led interventions and strategies on ED overcrowding and patient flow metrics. Although, what may work in one context may not work in another due to many factors (eg, funding, political views, type of healthcare systems, etc) that may play a role in determining the best intervention or strategy for a specific context. It is important to conduct systematic reviews to critically appraise and summarise evidence on the effectiveness of specific 'within ED' or 'outside ED' interventions involving PHCPs to mitigate ED overcrowding. In this scoping review, we only identified eight relevant randomized controlled trials (RCTs) that reported PHCP-led interventions. Thus, it is important to conduct well-designed RCTs evaluating the effectiveness of specific interventions involving PHCPs on ED outcomes to generate high-quality evidence on this topic.

This review has numerous strengths. To limit selection bias, multiple steps and multiple independent reviewers reviewed the studies to make selections in an unbiased manner. One of the major strengths of this scoping review is that we engaged and collaborated with 12 patient partners from multiple provinces across Canada during the design stage to refine our review question and identify patient-important review outcomes, as well as during the conduct stage and knowledge dissemination stages. In addition, the JBI guidelines were used for conducting this scoping review along with an a priori registered protocol. A comprehensive search strategy was developed by an experienced health librarian and peer reviewed by a second information specialist using PRESS guidelines along with search for grey literature sources, to identify relevant studies to answer our review question. In addition, the search strategy was reviewed by our patient partners who collaborated on the project. We did not have any restrictions on publication dates, population, or study design (with exception of narrative reviews, editorials, commentaries and historical articles).

There are some important limitations with this review to be considered. In our review, publication bias was only partially addressed. Despite using a comprehensive and exhausting search strategy, we made a pragmatic decision to only include English language publications. Consequently, it is possible that some of the studies from countries where English is not the first language may have been missed. Our review focused on the interventions and strategies that involved only PHCPs such as FPs, NPs and nurses with increased authority. Studies that involved ED physicians or specialists were not considered in this review for feasibility. We had also made an a priori decision, for feasibility, to not include studies that had exclusively investigated the role of physician assistants, but if a study reported them to be a part of a team that consisted of PHCPs, then we included those studies in our scoping review. We included studies that had involved any one of the included PHCPs individually or as a part of a team that had other professionals, and thus in these studies,

the impact of intervention cannot be solely ascribed to a specific PHCP.

## Conclusions

This review provides a timely contribution to improving the understanding of the growing problem of ED overcrowding. In this scoping review, we have identified a variety of interventions 'outside' and 'within' the ED involving PHCPs from the existing literature and we have summarised the nature of the impact of these interventions on 'input'-related (ED utilisation) and 'throughput'-related (ED patient flow metrics) outcomes. We have also identified and reported innovative ways in which PHCPs may be engaged in patient care. The geographic area in which these studies were conducted, and the type of healthcare systems vary widely across the included studies and may influence how these results can be interpreted. In the future, it will be important for researchers to focus on conducting high-quality systematic reviews to synthesise evidence on the effectiveness and unintended consequences of specific 'within ED' or 'outside ED' interventions involving PHCPs. It will also be important to engage diverse patient and caregiver perspectives in the systematic review process to align with patient-important priorities and key patient-identified outcomes prior to conducting the systematic reviews. Finally, researchers should be encouraged to conduct well-designed RCTs to generate high-quality evidence on this topic, as there have been a limited number of RCTs conducted on this topic.

**Author affiliations**
[1]George and Fay Yee Center for Healthcare Innovation, University of Manitoba, Winnipeg, Manitoba, Canada
[2]Department of Community Health Sciences, Rady Faculty of Health Sciences, University of Manitoba, Winnipeg, Manitoba, Canada
[3]Max Rady College of Medicine, Rady Faculty of Health Sciences, University of Manitoba, Winnipeg, Manitoba, Canada
[4]Department of Emergency Medicine, Faculty of Medicine and Dentistry, University of Alberta, Edmonton, Alberta, Canada
[5]Department of Family Medicine, Rady Faculty of Health Sciences, University of Manitoba, Winnipeg, Manitoba, Canada
[6]Department of Medical Oncology and Hematology, CancerCare Manitoba, Winnipeg, Manitoba, Canada
[7]Axe Santé des populations et Pratiques optimales en santé, Centre de recherche du CHU de Québec-Université Laval, Laval, Quebec, Canada
[8]HEC Pôle santé, Université de Montréal, Montreal, Québec, Canada
[9]Department of Emergency Medicine, Cité de la santé de Laval, Laval, Quebec, Canada
[10]WRHA Virtual Library, University of Manitoba, Winnipeg, Manitoba, Canada
[11]Manitoba College of Family Physicians, Winnipeg, Manitoba, Canada
[12]Patient and Public Engagement Collaborative Partnership, George and Fay Yee Center for Healthcare Innovation, Winnipeg, Manitoba, Canada
[13]The Alberta Primary and Integrated Health care Innovation Network, Edmonton, Alberta, Canada
[14]Community Health, Quality and Learning, Shared Health Manitoba, Winnipeg, Manitoba, Canada
[15]Manitoba Primary and Integrated Health care Innovation Network, Winnipeg, Manitoba, Canada
[16]Knowledge Translation Program, Unity Health Toronto, St Michael's Hospital Li Ka Shing Knowledge Institute, Toronto, Ontario, Canada
[17]Department of Emergency Medicine, Faculty of Medicine, University of Manitoba, Winnipeg, Manitoba, Canada
[18]School of Public Health, University of Alberta, Edmonton, Alberta, Canada

**Acknowledgements** We are very grateful to the Canadian Institutes of Health Research (CIHR) (NKS 158643), Manitoba Medical Services Foundation (# 8-2018-06) and Winnipeg Foundation (# 8-2018-06) for providing financial support for this project. Dr Rowe's research was supported by a Scientific Director's Grant (SOP 168483) from CIHR. Dr Tricco's research is funded by a tier 2 Canada Research Chair in Knowledge Synthesis. We thank the Emergency Medicine Research Group in the Department of Emergency Medicine at the University of Alberta for in-kind resources. We are also very grateful to all the patient partners who collaborated with us in the design, conduct and dissemination stages of this project. We are thankful to Frank Krupka for the kind support of this project during his time as the executive director of CHI. We are very grateful to Dr Cristina Villa-Roel for the kind support of this project. We thank Tamara Rader, Christine Nielson and Janet Joyce for help with the search strategy for this study.

**Contributors** MMJ, AMA-S, TBe and MH substantially contributed to the design and conception of the study. MMJ drafted the manuscript with input from all co-authors. MMJ, AMA-S and RR were involved in the data analysis. NA was involved in developing search strategy. LC, RNA, YA-Y, SWK and NA-Y were involved in study selection process and in data extraction. MBD, SB, JM, PT, BHR, AC and WS provided content expertise in emergency medicine. RS, GH, TBu and JE provided content expertise in family medicine. CS provided content expertise in patient engagement. ACT, RZ and AMA-S provided content expertise in scoping review methodology. All study authors provided critical feedback and approved the final manuscript.

**Funding** This study was supported by th research grant from Canadian Institutes of Health Research (NKS 158643), Manitoba Medical Services Foundation (# 8-2018-06) and Winnipeg Foundation (# 8-2018-06).

**Competing interests** None declared.

**Patient consent for publication** Not required.

**Provenance and peer review** Not commissioned; externally peer reviewed.

**Data availability statement** All data relevant to the study are included in the article or uploaded as supplemental information.

**ORCID iDs**
Maya M Jeyaraman http://orcid.org/0000-0002-1548-3987
Malcolm B Doupe http://orcid.org/0000-0002-6889-9097
Gayle Halas http://orcid.org/0000-0003-0433-0632
Andrea C Tricco http://orcid.org/0000-0002-4114-8971

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
