## [Reviewer comments · BMJ Open]

ARTICLE DETAILS

TITLE (PROVISIONAL)	Interventions and strategies involving primary healthcare professionals to manage emergency department overcrowding: a scoping review
AUTHORS	Jeyaraman, Maya; Copstein, Leslie; Al-Yousif, Nameer; Alder, Rachel; Kirkland, Scott; Al-Yousif, Yahya; Suss, Roger; Zarychanski, Ryan; Doupe, Malcolm; Berthelot, Simon; Mireault, Jean; Tardif, Patrick; Askin, Nicole; Buchel, Tamara; Rabbani, Rasheda; Beaudry, Thomas; Hartwell, Melissa; Shimmin, Carolyn; Edwards, Jeanette; Halas, Gayle; Sevcik, William; Tricco, Andrea; Chochinov, Aleks; Rowe, Brian; Abou-Setta, Ahmed

VERSION 1 – REVIEW

REVIEWER	Rasouli, Hamid Baqiyatallah University of Medical Sciences
REVIEW RETURNED	18-Jan-2021

GENERAL COMMENTS	Please explain the necessity of doing this study by scoping review method in the introduction section. The conclusion was presented very general and it is not very informative. Please provide a conclusion based on the results. Please provide a table that lists information regarding the type of study, participants, interventions, and the effects and remove table 2 and Appendix Table 6. Please discuss more the mechanism of intervention and discuss disagreements outcomes and negative effects too.
--

REVIEWER	Cabilan, Cara Queensland Health
REVIEW RETURNED	28-Jan-2021

GENERAL COMMENTS	Well written systematic scoping review: good justification, clear methods. The involvement of consumers is commendable. My feedback mainly surrounds the reporting of results. As a researcher in the emergency department, I would like to have a clear picture of where research is needed. After all, that is what scoping reviews are for. In this vein, I would have liked to know: 1) presenting complaints that can be managed by PCHPs2) sample sizes3) which PHCP interventions tend to be more impactful In addition, to provide clarity on the outcomes, perhaps the outcomes can be categorised such as input, throughput, output, or other. Another way to do this would be to group outcomes
---

	according to quality of care domains: safety, timeliness, effectiveness, efficiency, patient-centredness, and equitability. A minor point: reference section needs attention, inconsistencies with capitalisation of journal names, abbreviation, formatting, etc.
--	--

VERSION 1 – AUTHOR RESPONSE

Author response to comments from the reviewers:

Reviewer 1

1. Please explain the necessity of doing this study by scoping review method in the introduction section.

Authors' response: Thank you. Our apologies for not being clear. As suggested, we have changed the last paragraph of the introduction section to explain why we chose the scoping review methodology. See edits in bold below:

As such, it is crucial to identify interventions and strategies involving primary healthcare professionals (PHCP) that have an impact on the ED metrics. Using a scoping review methodology, in order to map the existing literature on this topic, the main objective was to identify and summarize existing literature on the interventions and strategies involving PHCPs (family physicians/general practitioners (GP), nurse practitioners (NP), or nurses with expanded role) to manage ED overcrowding.

2. The conclusion was presented very general and it is not very informative. Please provide a conclusion based on the results.

Authors' response: Thank you. As suggested, we have made changes to our conclusion section to reflect the results of our review. See edits in bold below:

This review provides a timely contribution to improving the understanding of the growing problem of ED overcrowding. In this scoping review, we have identified a variety of interventions "outside" and "within" the ED involving PHCPs from the existing literature and we have summarized the nature of the impact of these interventions on "input" (ED utilization) and "throughput" (ED patient flow metrics) related outcomes. We have also identified and reported innovative ways in which PHCPs may be engaged in patient care. The geographic area in which these studies were conducted, and the type of healthcare systems vary widely across the included studies and may influence how these results can be interpreted. In the future, it will be important for researchers to focus on conducting high quality systematic reviews to synthesize evidence on the effectiveness and unintended consequences of specific "within ED" or "outside ED" interventions involving PHCPs. It will also be important to engage diverse patient and caregiver perspectives in the systematic review process to align with patient-important priorities and key patient-identified outcomes prior to conducting the systematic reviews. Finally, researchers should be encouraged to conduct well-designed RCTs to generate high quality evidence on this topic, as there have been a limited number of RCTs conducted on this topic.

3. Please provide a table that lists information regarding the type of study, participants, interventions, and the effects and remove table 2 and Appendix Table 6.

Authors' response: Thank you. As suggested, we have added two tables (in the Appendix- Appendix table 7 & 8) listing all the information you have requested. We were unable to add these new tables to the main manuscript as both the new tables were very long (>20 pages each) as it contains details from each of the 268 included studies. Also, as suggested, we have removed Table 2.

4. Please discuss more the mechanism of intervention and discuss disagreements outcomes and negative effects too.

Authors' response: Thank you. We appreciate your suggestion. Our study is a scoping review, and our objective was to extract and report on all the available interventions involving primary healthcare providers (PHCPs) and their reported impact on the ED outcomes, from the current literature. We agree that the mechanisms of the various identified interventions and their individual impacts are important, but these details are beyond the scope of our current review, and we are considering it in our future work.

Reviewer 2

Comments to the Author:

1. Well written systematic scoping review: good justification, clear methods. The involvement of consumers is commendable.

Authors' response: Thank you.

2. My feedback mainly surrounds the reporting of results. As a researcher in the emergency department, I would like to have a clear picture of where research is needed. After all, that is what scoping reviews are for. In this vein, I would have liked to know:

- 1) presenting complaints that can be managed by PHCPs
- 2) sample sizes
- 3) which PHCP interventions tend to be more impactful

Authors' response: Thank you. We appreciate your suggestions.

As a part of our scoping review, we did include the sample size of each study as one of the variables to extract, but unfortunately, we found that most of the included studies did not report this information. Hence, we were unable to provide details on sample size in the manuscript. Regarding your question on which PHCP interventions are more impactful, we have added a new table in the appendix section (Appendix table 8) based on request from reviewer 1, that reports the nature of impact of the interventions for each of the 268 included studies separately. Lastly, we agree that the presenting complaints that can be managed by the PHCPs are important, but extraction of this information is beyond the scope of our current review and we will consider it in our future work.

3. In addition, to provide clarity on the outcomes, perhaps the outcomes can be categorised such as input, throughput, output, or other. Another way to do this would be to group outcomes according to quality of care domains: safety, timeliness, effectiveness, efficiency, patient-centeredness, and equitability.

Authors' response: Thank you for your suggestion. We agree these are important ways to categorize outcomes. We have provided the list of outcomes reported by each included study under Appendix table 8. In this review we mainly report the input and throughput outcomes. The "outside ED" interventions mainly reported the "input" outcomes (ED utilization) and the "Within ED" interventions reported "throughput" outcomes (ED patient flow metrics). We have added this statement in the results and discussion section to clarify this point as shown in bold below:

Results:

The majority of the included studies that reported "outside ED" interventions or strategies investigated the impact of these interventions on "input" related outcomes (i.e., minimizing the ED utilization), whereas studies reporting "within ED" interventions or strategies investigated the impact of the interventions on "throughput" related outcomes (i.e., ED patient flow metrics).

Discussion:

It is important to note that studies reporting "outside ED" interventions and strategies focused predominantly on reduction of ED utilization by patients and reported "ED visits" as an outcome (input related outcomes) in their study report. Although we are unaware of any scoping reviews on "outside ED" PHCP-led interventions, there have been a few focused systematic reviews published to date on specific areas under this broad topic.

3. A minor point: reference section needs attention, inconsistencies with capitalisation of journal names, abbreviation, formatting, etc.

Authors' response: Thank you. We have made the suggested changes and updated the references.

VERSION 2 – REVIEW

REVIEWER	Rasouli, Hamid Baqiyatallah University of Medical Sciences
REVIEW RETURNED	01-Apr-2021

GENERAL COMMENTS	Accept
--------

REVIEWER	Cabilan, Cara Queensland Health
REVIEW RETURNED	19-Apr-2021

GENERAL COMMENTS	Thank you for the revisions. Feedback had been carefully considered and revisions were appropriate.
---